# In Vitro Models of Bone Marrow Remodelling and Immune Dysfunction in Space: Present State and Future Directions

**DOI:** 10.3390/biomedicines10040766

**Published:** 2022-03-24

**Authors:** Ryan Sarkar, Francesco Pampaloni

**Affiliations:** Buchmann Institute for Molecular Life Sciences (BMLS), Johann Wolfgang Goethe Universität, 60438 Frankfurt am Main, Germany; sarkar@bio.uni-frankfurt.de

**Keywords:** 3D cell culture, bone marrow niche, hematopoiesis, hematopoietic progenitor cells, innate immunity, mesenchymal stem cells, microgravity, myelopoiesis

## Abstract

Spaceflight affects the body on every level. Reports on astronaut health identify bone marrow remodelling and dysfunction of the innate immune system as significant health risks of long-term habitation in space. Microgravity-induced alterations of the bone marrow induce physical changes to the bone marrow stem cell niche. Downstream effects on innate immunity are expected due to impaired hematopoiesis and myelopoiesis. To date, few studies have investigated these effects in real microgravity and the sparsely available literature often reports contrasting results. This emphasizes a need for the development of physiologically relevant in vitro models of the bone marrow stem cell niche, capable of delivering appropriate sample sizes for robust statistics. Here, we review recent findings on the impact of spaceflight conditions on innate immunity in in vitro and animal models and discusses the latest in vitro models of the bone marrow stem cell niche and their potential translatability to gravitational biology research.

## 1. Hematopoiesis, Innate Immunity, and Spaceflight Conditions

The rapid pace of technological advancements in spaceflight has transformed the dynamic such that the physiology of the human body and its ability to withstand the long-term effects of space travel is now one of the greatest factors precluding further exploration into space [1,2]. Space travel presents individuals with numerous physical and mental challenges, as shown in Figure 1. Microgravity, the condition of apparent weightlessness, is a major environmental factor in space and exposes astronauts to physiological stress stemming from reduced mechanical loads on the weight-bearing structures of the body and a redistribution of bodily fluids [3]. Further health risks are posed by increased exposure to radiation (especially relevant for missions beyond low Earth orbit) and psychosocial changes from travel-associated long-term confinement [3,4].

Given the demands of space travel, preserving the immune health of astronauts is of utmost importance. The immune system consists of two interlinked subsystems, the slow but specific adaptive immune system and the rapid, broadly responding innate immune system. The delayed response of adaptive immunity is antigen-specific, a trait resulting in the system maintaining a long-term memory [5]. In contrast, the innate immune system is non-specific and represents the first line of defence against pathogens, with key functions including the destruction of pathogens, induction of inflammation, and activation of the adaptive immune response [6,7]. This system relies on phagocytic cells recognizing conserved features of pathogens enabling their quick activation, as well as the release of cytokines and chemokines to facilitate immune cell recruitment [6]. That the adaptive and innate immune system are affected by microgravity and radiation has been well established, but the cellular mechanisms of long-term exposure are only beginning to be investigated [5,8,9,10,11,12,13].

Along with immune dysfunction, astronauts also experience bone marrow density loss and remodelling while in space, with the current literature suggesting these two issues are linked [5,14]. Accordingly, alterations to the hematopoietic stem cell (HSC) niche, located in the bone marrow, play a primary role. The HSC niche is crucial in the establishment and maintenance of innate immunity (Figure 2), and as such is likely a central aspect of any potential links between microgravity-induced bone marrow loss and the immune dysfunction experienced by astronauts [5,14,15,16,17,18]. Hematopoietic stem cells give rise to all blood cells through hematopoiesis. Consequently, bone marrow remodelling related to long-term exposure to microgravity and the associated impacts on hematopoietic function can lead to severe health consequences [19]. Yet the mechanisms of the relationship between the bone marrow remodelling in microgravity, its effect on hematopoietic progenitor cells, and immune dysfunction remains unknown. Studies performed on astronauts to further elucidate these relationships have been limited by their small sample size. While research on microgravity, radiation, and the immune system has also been performed on animal models, these too are often limited by small sample size. Furthermore, the reactions of animals are often not representative of human physiology, and their use precludes downstream applications in humans, thus further emphasizing the need for cellular models [8,20,21].

The complexity and limitations of in vivo studies in real microgravity demonstrate that the development of high-throughput in vitro models, physiologically relevant to humans, and capable of delivering appropriate sample sizes for robust statistics are necessary. Such models would increase the understanding of the mechanisms of microgravity adaptation on a cellular level. Additionally, the use of cellular models as opposed to studies of astronauts facilitates analysis via high-throughput techniques such as immunofluorescence microscopy as well as the screening of drugs mitigating the detrimental effects of microgravity.

## 2. Response of the Innate Immune System to Spaceflight Conditions

The previous decades of space exploration have provided a wealth of evidence associating spaceflight with immune system dysfunction [5]. Studies on astronauts have demonstrated that neutrophil count, phagocytic function, and oxidative function are affected by spaceflight [22,23,24]. Other studies reported that long duration spaceflight impairs function of Natural Killer (NK) cells in astronauts [25]. In a study of 11 astronauts to fly on the space shuttle, 7 were found to have reduced levels of monocytes [26]. Consequently, microgravity-induced immune dysfunction currently presents one of the greatest barriers to long-distance space travel, affecting both the adaptive and innate immune systems comprehensively [5]. As most cells of the innate immune system do not divide but are instead the product of hematopoietic stem cells (Figure 2), increasing the present understanding of the impact of microgravity on hematopoiesis remains of utmost importance.

### 2.1. Modelling Microgravity

Platforms to study microgravity can be broadly divided into two categories, real and simulated. Real microgravity platforms include drop towers, parabolic flights, sounding rockets, and orbital platforms, with each facilitating increasingly longer durations of experimentation. The various platforms have their own advantages and drawbacks, but the only facility able to sustain long-term exposure to microgravity is also by far the most expensive, the orbital platform [27].

The challenges and cost associated with studying real microgravity led to the development of devices designed to simulate it. These devices exploit the delay in organisms sensing gravity, as well as the time taken for sedimentation, to prevent samples from determining the true gravitational vector [27]. As such, the degree to which gravity is simulated depends both on the threshold of sensitivity a specimen has to gravity and on the devices and programs used in the simulation [28]. Devices designed for simulating microgravity include Rotating Wall Vessels (RWV) and clinostats which function by rotating at a constant speed, ensuring the samples within are in a constant state of free fall, as shown in Figure 3. The Rotating Wall Vessel, sometimes referred to as the Rotary Cell Culture System (RCCS), rotates around one axis, as does a 2D clinostat. The RWV generally rotates faster than 2D clinostats to counteract sample sedimentation and thus is well-suited for suspension cultures [27]. In contrast to these devices, 3D clinostats and the Random Positioning Machine (RPM) rotate on two axes. The 3D clinostat rotates at a constant speed and direction while the RPM is comprised of two independent frames that rotate at different speeds and in different directions from each other. The foundational principle of the RPM differs from the other devices that place samples in a constant state of free fall; instead, it aims to distribute the gravitational vector evenly in all directions so that the gravitational vector experienced by the sample averages over time to zero [29,30]. Centrifugal forces increase with distance from the centre of rotation and, as all these platforms function via rotation, the most effective simulation of microgravity sees the samples as close to the centre of rotation as possible to minimize the effect of fluid dynamics and centrifugal forces [31]. The speed of rotation is of great importance as it must be rapid enough to prevent sedimentation, yet slow enough to prevent centrifugal forces large enough to trigger the gravitational sensitivity threshold of the systems being studied, and in the case of clinorotation, the speed must also be constant [32]. Faster rotation entails greater centrifugal forces and thus higher rates of rotation create a smaller effective radius from the centre of rotation for valid simulations [31]. Accordingly, rapidly rotating 2D clinostats can typically only support small specimens in contrast to the large sample volumes supported by slower rotating platforms such as the RPM. However, the RPM also has limitations such as high shear stresses in certain applications [30,33]. Additionally, the random positioning algorithms have been observed to produce results differing from those obtained in real microgravity, in contrast to results obtained when the RPM was run under 3D clinorotation [34]. Furthermore, the time-averaged nature of the RPM requires it to operate for a longer time in order for the simulation of microgravity to be as intended. This necessity may preclude its use for experimental systems that are quick to sense gravity [33].

One of the key benefits of simulated microgravity is the considerable reduction in the cost of entry to space-related research. By democratizing microgravity, simulated microgravity allows researchers to perform experiments that would otherwise remain inaccessible to them. Additionally, simulated microgravity allows the usage of complex biological systems and high sample sizes that would otherwise be unfeasible under the constraints of orbital experiments. Accordingly, ground-based devices are often used as a preliminary step to determine whether a biological process is sensitive to gravity. However, results obtained from different simulated microgravity platforms often report conflicting results. This phenomenon may stem from specific artefacts inherent to each platform, such as centrifugal accelerations or vibrations in the case of clinorotation [35]. With ground-based devices, the potential for artefacts affecting the integrity of simulation requires validation of the experimental setup. However, proper validation necessitates direct comparison of results obtained from ground-based microgravity systems with data obtained in real microgravity [35]. Yet, such a workflow reintroduces the issue of a lack of access to real microgravity that these platforms were meant to address, and as such is often not feasible. In lieu of real microgravity for validation, perhaps variance in results could be addressed by larger-scale studies using multiple simulation platforms. Enabling additional comparisons between different systems may address the question of artefacts of any one system impacting study results. An additional issue inherent to experiments using a single platform has been demonstrated by Chapes and Ortega. Using identical cells, they introduced differences in the cell culture device and orientation of rotation of RWVs and noticed strong effects on the outcomes [36]. They recovered more viable cells and more differentiated cells when culture vessels were rotated horizontally as opposed to vertically. Additionally, the smaller culture vessels yielded more viable cells than larger bioreactors. Their findings highlight the importance of ground-based studies to determine the impact of a cell culture system on an experiment and ensure accurate interpretation of space flight data [36]. Taken together, while real microgravity is essential for validating the suitability of ground-based platforms, simulated microgravity also proves invaluable in preparation for space studies [35,36].

### 2.2. Response of Innate Immune Cells to Microgravity and Ionizing Radiation

#### 2.2.1. Hematopoietic Stem Cells

The study of hematopoiesis, whether on Earth or in space, revolves around discussion of hematopoietic progenitor cells. Crucial characteristics of progenitor cells are their self-renewal and their capacity to differentiate into diverse types of cells with diverse functions. As all types of immune cells originate from hematopoietic progenitor cells, damage linked to their integrity can affect all downstream immune responses.

The capacity of HSC to proliferate and differentiate has been widely studied. Early studies investigating the mechanisms and signalling pathways involved in HSC proliferation and differentiation under microgravity showed that CD34+ HSC (CD34 being a typical marker of HSC progenitors) maintained at 1 *g* proliferate three-fold faster than HSC exposed to simulated microgravity in a RWV [37]. In fact, HSC under microgravity left the G_0_/G_1_ phase slower than the cells at 1 *g*. Furthermore, the hematopoietic potential of the HSC in microgravity was higher and more persistent in time compared to the 1 *g* condition. However, this study did not investigate the molecular mechanism at the base of the observed phenotype. A follow-up study by the same group found that following three days of culture in the RWV, HSC downregulated the expression of stromal cell–derived factor 1 (SDF-1α) and F-actin [38]. This induces a reduction in the migratory ability of HSC that correlates with a reduced tendency of the cells to differentiate to mature progenitors. These results provided a first mechanistic explanation of the effect of microgravity on HSC proliferation and differentiation through HSC mobilization pathways.

In another experiment featuring two space flights, Wang et al. found murine HSC to have proliferated significantly less and express significantly lower levels of the proliferation marker Ki67 following 12 days of spaceflight in comparison to ground controls [39]. This was also observed when the experiment was repeated in simulated microgravity using an RWV, with the cell cycle primarily being blocked at the G1/S transition. A similar study performed on human CD34+ HSC isolated from adult bone marrow samples showed decreased cell proliferation in a RWV when compared to 2D controls, although in this case the primary point of cell cycle arrest was the G2/M phase [40]. The results of both studies coincide with those of Davis et al., who found human HSC in co-culture with endothelial cells to exhibit a 57–84% decrease in proliferation after 11–13 days of real microgravity exposure aboard the Space Shuttle in comparison to ground-based controls [41]. Additionally, different cell types had different responses to microgravity with myeloid progenitors expanding less in space than in normal gravity while the erythroid progenitor population in the space shuttle actually shrank in contrast to the expansion seen in ground controls [41].

Experimental evidence from previous studies demonstrates that mechanical forces play a crucial role in determining HSC fate [42,43]. HSC lineage differentiation depends on the stiffness of the surrounding microenvironment: HSCs in contact with a soft substrate (0.71 KPa) display a round shape, lack of polarity in the actin cytoskeleton, and quiescence, while HSCs cultured on a stiff (196 KPa) matrix show formation of cell protrusions, cell polarity, and fast decision to a specific cell fate [44]. From these results, it follows that the mechanical unloading sensed by HSC under microgravity, mediated by the actin cytoskeleton, would impair their polarization, migration ability, and proliferation potential [44]. Microgravity-responsive genes were identified at a systemic level by RNAseq transcriptomics analysis of murine HSC cultured on the Tianzhou-1 orbital cargo ship over 12 days [39]. GO-biological process enrichment analysis found that genes belonging to apoptosis, hypoxic stress, hematopoietic capacity, and inflammatory response processes were upregulated. In contrast, a downregulation of cell proliferation pathways and blockage of the G_0_/G_1_ phase were detected, in agreement with Plett et al., 2001 [37,39]. Notably, the expression of the c-KIT SCF (Stem Cell Factor) receptor, involved in HSC quiescence and maintenance, was significantly downregulated under microgravity. SCF is a pivotal cytokine promoting proliferation, differentiation, and migration of HSC, which activates HCS proliferation through the PI3K/AKT and MEK/ERK [39,45]. Interestingly, the ERK signalling is one of the main regulators of cell motility [46]. Thus, perturbation of the ERK pathway is a possible mechanism linking mechanical unloading of bones under microgravity and the low proliferation and differentiation of HSC.

An ISS experiment on human blood-derived stem cells determined that microgravity results in a more pronounced loss of pluripotency compared to controls on the ground, as well as increased osteogenic differentiation [47]. Conversely, RWV culture of human CD34+ HSC cells found simulated microgravity to inhibit differentiation when compared to static control cultures [40]. The origin of the cells may also play a role, with Davis et al. observing human HSC in space to differentiate towards the macrophage lineage, while Wang et al. observed no microgravity-mediated effects on differentiation in murine HSC [39,41]. As stated earlier, previous research has highlighted biomechanical forces as important factors to consider in differentiation [43]. A study on embryonic hematopoiesis demonstrated the importance of shear stress and mechanical loading in HSC differentiation, maturation, and colony-forming potential—such factors would be affected in space studies, such as those of Wang et al. and Davis et al., as fluid dynamics and mechanical forces are strongly reduced in microgravity when compared to 1 *g* [42].

The typical microgravity HSC phenotype consisting of low proliferation and low differentiation directly affects the innate immune system. In fact, the circulating immune cells have a short life span and a continuous and sustained replenishment of blood immune cells is needed for a healthy immune system [48]. The perturbation of this homeostatic mechanism by microgravity leads to an immediate, albeit transient, effect on the innate immune response

Collectively, the effects of microgravity on HSC generally involve a decrease in the rate of proliferation and population expansion of progenitor cells. Differentiation has also observed to be affected both in real and simulated microgravity. Given the cell types descended from HSC, these effects could cause the impairment of the innate system function in astronauts.

#### 2.2.2. Peripheral Blood Mononuclear Cells

The peripheral blood mononuclear cells (PBMC) are the main immune cell group isolated from the blood cells and comprise cells such as Natural Killer (NK) cells, dendritic cells, monocytes, lymphocytes, and macrophages. These cells possess a round nuclei in comparison to the multi-lobed nuclei found in granulocytes.

Moreno-Villanueva et al. subjected the entire PBMC population to varying degrees of ionizing radiation while incubated in an RWV [49]. Although increased background radiation is another major environmental factor of space travel, very few studies have examined the effect of simultaneous microgravity and radiation exposure. Ionizing radiation, such as that used in this experiment, is what humans are generally exposed to on Earth. Radiation from beyond Earth’s orbit is quite different, however, as it contains high-energy protons, solar particles, and charged particles [1]. Unfortunately, current facilities are unable to replicate this nonionizing radiation on Earth, limiting its study to space. Accordingly, studies such as this one by Moreno-Villanueva et al. are important for providing a preliminary understanding of the biological effects of spaceflight conditions, despite being limited to ionizing radiation.

When PBMC were subjected to an absorbed exposure of 0.8 Gy radiation, the rate of apoptosis significantly increased in 1 *g* controls but, interestingly, no change was observed in cells in simulated microgravity. Additionally, radiation induced more double-strand breaks in RWV cells. Radiation was only seen to induce cytokine release under simulated microgravity, which it did in a dose-dependent manner, with higher radiation inducing a greater cytokine release. To mitigate the combined effects of microgravity and radiation, the authors treated the cells with the sympathomimetic drug isoproterenol. This treatment would induce the release of stress hormones which influence immune regulation, DNA repair, and bone homeostasis, thereby reducing the effects of microgravity [49]. Based on their results, the authors concluded that isoproterenol did prevent most microgravity-mediated effects. In contrast, at a level of 2 Gy, radiation isoproterenol treatment was much less effective in cells incubated in the RWV compared to 1 *g* controls.

As space is characterized not only by microgravity but also elevated background radiation, this study is valuable in establishing a knowledge base into the combined effects of radiation and microgravity on PBMC.

#### 2.2.3. Monocytes and Macrophages

Of the multiple cell types comprising PBMC, monocytes, and their descendants, macrophages, are some of the most widely studied. Non-polarized M0 Macrophages can be polarized into either the pro-inflammatory M1 phenotype or the pro-healing/anti-inflammatory M2 phenotype [50]. The three phenotypes have been shown to have differing responses to simulated microgravity. A study by Ludtka, Moore, and Allen reported a RWV clinorotation-mediated shift from the initial singular phenotypes to mixed populations of cells co-expressing M1 and M2 specific genes [50]. Another study using murine HSC progenitors found that both spaceflight and RWV-simulated microgravity reduced macrophage differentiation from HSC progenitors, the quantity of macrophages, and their functional polarization [51]. Further analysis with qPCR revealed that key genes relating to macrophage proliferation and differentiation were downregulated in both spaceflight and RWV-simulated microgravity. Interestingly, stimulation of the RAS, ERK, and NFκB signalling pathways of cells cultured in the RWV was found to partially rescue the effects of simulated microgravity, perhaps suggesting potential targets for space medicine. Further research into the underlying mechanisms of real and simulated microgravity on the macrophage phenotype could elucidate the effect of microgravity on innate immunity.

Reporting on phenotypic changes, researchers have conducted experiments on primary human macrophages in real microgravity on sounding rockets. The macrophages have been observed responding to microgravity in a matter of seconds, exhibiting cytoskeletal changes in addition to increases in the volume and surface area of both cells and nuclei [52]. However, following the initial swelling period, cell volume and surface area shrank significantly below the starting values. Despite the shrinking, upon return to Earth, no structural changes were observed, indicating a recovery of the original cytoskeletal organization.

The structure of macrophages plays an important role in determining their function, with specific defined features corresponding to their status [53]. M1 macrophages, for example, are typified by lamellipodia and filopodia, with the actin cytoskeleton distributed throughout the cell [53]. In contrast, M2 macrophages exhibit a rounded shape, with actin concentrated around the nucleus [54]. Questions over whether structural changes in microgravity affect macrophage function have led to studies examining their oxidative burst reaction. Notably, 2D clinostat-simulated microgravity reduced ROS production in NR8383 rat macrophages, while parabolic flight resulted in ROS release to increase and decrease in hyper- and microgravity, respectively [55,56]. These results indicate a gravisensitive step in signalling, ultimately identified as Syk phosphorylation [56]. Other studies demonstrated that NR8383 cells adapt to real microgravity within 1 min, but also rapidly re-adapt to 1 *g* conditions [57]. This adaptability was also seen in human cells during longer-term research aboard the ISS, which found primary human macrophages to exhibit no structural changes in the actin and vimentin cytoskeletons after 11 days of spaceflight when compared to 1 *g* controls [58]. The transient nature of the structural changes in macrophages and the rapid cellular response observed in these studies suggest the establishment of an adapted steady state by the macrophages based on their gravitational condition.

In contrast, U937 myelomonocytic cells exhibited decreased proliferation, actin expression, and cytoskeletal disorganization after 72 h of RWV culture [59]. These results coincide with those of Paulsen et al., where U937 cells exhibited disruption of the actin cytoskeleton and disorganization of tubulin following a five-day spaceflight [60]. These cells also had reduced expression of CD18, CD36, and MHC-II. Such a phenotype, the authors report, would be incapable of migration, recognizing pathogens, attacking pathogens, or activating the adaptive immune system. Further research found macrophage-like differentiated U937 cells to express higher levels of intracellular adhesion molecule 1 (ICAM-1) in 2D clinostat-simulated microgravity, parabolic flight, and spaceflight [61]. These effects were also observed in primary human M2 macrophages yet undifferentiated U937 showed no microgravity-related changes in expression of ICAM-1. Conversely, spaceflight was found to decrease the levels of ICAM-1 expression in human primary M1 macrophages, potentially hindering the migration ability of the cell [58]. These varying results could arise from differences in macrophage polarization or be cell-type dependent. A study by Moser et al. found that in microgravity during parabolic flight, PBMCs close to the ICAM-1 coated substrate were moved to the centre of the chamber, thus rendering binding impossible and likely preventing immune activation [62]. These results raise the additional question as to what extent decreased ICAM-1 expression impairs the ability of the cell to migrate in microgravity versus an inability to even reach a substrate.

Finally, osteocytes present an additional factor to consider as a result of the role they play in macrophage activation. Representing more than 90% of the bone cell population, osteocytes can sense and transmit mechanical stimuli over long distances through the bone via their elongated cell processes, which build an extended network of cell–cell and cell–matrix contacts with other bone cell types and the bone extracellular matrix. Osteocytes react to mechanical cues by releasing inflammatory mediators including nitric oxide and prostaglandin E_2_, which favours macrophage maturation and activation [63]. Compelling evidence suggests that macrophages are pivotal cells for the maintenance of the bone marrow HSC niche and for the modulation of osteogenesis [64]. Activated (M1) macrophages induce bone resorption by releasing the pro-inflammatory cytokines TNFα, IL-1β, and IL-6. IL-1 induces bone loss by the NF-kB/RANKL pathway-induced osteoclast formation. IL-6 activates the osteocyte-mediated osteoclasts’ formation through the JAK2/RANKL pathway [65].

#### 2.2.4. Osteocytes

Studies aimed at understanding the modulation of the hematopoietic microenvironment by bone cells, a vital research field known as osteoimmunology, found that osteocytes (which represent 90–95% of all the bone cell mass) and the bone extracellular matrix are closely interconnected with the HSC niche [66]. Osteocytes control the HSC through both soluble factors and direct cell–cell communication with HSC via intermediate cells of the stem cell niche, such as osteoblasts and osteoclasts [67]. Osteocytes are mechanosensory cells and sense the shear and compressive forces in the bone extracellular matrix, such as the mechanical unloading occurring in microgravity. Microgravity impairs the formation of the osteocyte network and subsequently reduces the level of osteocyte-secreted Granulocyte Colony-Stimulating Factor (G-CSF) [68]. G-CSF directly stimulates the production of leukocytes from the HSC niche, therefore, low levels would be concordant with immune dysfunction [68].

#### 2.2.5. Natural Killer Cells

Natural killer (NK) cells are key effectors of the innate immune system mediating anti-tumour and anti-viral responses [69]. Spaceflight is known to impair the anti-viral capabilities of the immune system, with over half of all astronauts demonstrating herpes virus reactivation suggesting microgravity-induced NK cell dysfunction [25,70]. Spaceflight has also been shown to significantly reduce the number and proportion of NK cells in astronauts, although levels were observed to recover within 10 days after landing [71].

Short-term experiments aboard the ISS with primary NK cells and K562 tumour cells in co-culture found the level of NK cytotoxicity and interferon production to be unaffected by microgravity [72,73,74]. Microgravity simulated by a 2D clinostat was also observed to have no effect, leading the authors to conclude that microgravity bears no effect on critical NK function. In contrast, Li et al. performed a study of ex vivo expanded primary NK cells in a horizontally-rotating RWV and found cytotoxicity to significantly decrease after 48 h and even further after 72 h when compared to 1 *g* and vertical rotation controls [75]. Additional analysis of these cells observed increased apoptosis, necrosis, decreased secretion of IFN-γ and perforin, and downregulated cell surface receptor expression to be potential contributors to the loss of cytotoxicity. Interestingly, after removal from the RWV, NK cells recovered quickly, within only three days at 1 *g*. Another study observed NK cell viability dropping significantly after only 12 h in a RWV [76]. This discrepancy could originate from differences in cell type but also from the difference in culture conditions. Li et al. used NK cells expanded ex vivo in monoculture in the RWV compared to the freshly harvested cells of Mylabathula et al., where all PBMC were cultured together in the RWV with the NK cells being isolated afterwards [75,76]. More recent work has suggested that downregulation of the NKG2D receptor induced by simulated microgravity is the cause of this NK cell dysfunction [77]. Contrary to the work of Li et al., following exposure to RWV-simulated microgravity, Mylabathula et al. observed no change in the cell surface phenotype of NK cells, though they acknowledge this may be due to the limited length of the experiment necessitated by the cytokine-free culture [75,76]. In both works, the NK cells and K562 cells were cultured together in 1 *g* after exposure to simulated microgravity to facilitate contact between the cells, but as noted by Mylabathula et al., this potentially allows the NK cells the chance to recover from their treatment. Although not completely in concordance, both papers provide insight into the factors contributing to the impaired function of NK cells in microgravity environments.

#### 2.2.6. Granulocytes

Granulocytes are a family of white blood cells including neutrophils, eosinophils, basophils, and mast cells. The latter three play key roles in allergic inflammation while neutrophils, also called polymorphonuclear leukocytes due to their lobed nuclei, are noted for their immediate and non-specific destruction of invading pathogens [78,79]. Research on microgravitational effects in granulocytes is lacking with the few published works investigating neutrophils.

A study by Paul et al. used the Neutrophil-to-Lymphocyte ratio (NLR) as a marker to determine the status of the immune system in experiments performed in real and simulated microgravity as astronauts had previously demonstrated elevated granulocyte-to-lymphocyte ratios [80]. Treatment in the RWV increased the NLR and the number of granulocytes in human leukocytes in vitro, in concordance with what the authors observed when examining previous data from experiments on astronauts. RWV-simulated microgravity also elevated the release of ROS and induced neutrophil activation. Notably, these effects could be reduced by treatment with the antioxidant N-acetyl cysteine. In contrast, an earlier study found that RPM-simulated microgravity does not affect the oxidative burst reaction in neutrophils, though this discrepancy could stem from the differing experimental platforms [81].

### 2.3. Response to Microgravity of the Innate Immune System in Animal Models

In addition to in vitro experiments, numerous animal experiments have also been flown to space to study the effects of microgravity on the immune system in vivo. Mice represent one of the most commonly utilized animal models in space experiments. In 2013, Blaber et al. reported significant bone resorption of the trabecular endosteal surface in the femoral head of mice after 15 days of spaceflight, thereby enlarging the bone marrow cavity [82]. Cells isolated from the femoral head also exhibited reduced expression of markers for early hematopoietic and mesenchymal differentiation. The differentiation potential of bone marrow MSC (mesenchymal stromal progenitor cells) was found to increase upon reloading, while microgravity was found to predispose HSC progenitors to differentiate towards the osteoclast lineage upon reloading at 1 *g*, with the authors suggesting an accumulation of undifferentiated progenitors occurring as a result of exposure to microgravity. That stem cell differentiation is affected by microgravity was also observed in in vitro experiments [41,47]. Blaber et al. suggest that a reduction in mechanical load, as experienced in microgravity, may inhibit the differentiation of stem cells, potentially leading to detrimental effects on regenerative health [82]. While this study provides valuable insight into bone remodelling under microgravity, it is important to note that only eight mice were used.

Much like the devices used to simulate microgravity for cell cultures, platforms also exist for the simulation of microgravity for animal experiments. Recently, the European Space Agency has unveiled a Random Positioning Machine able to subject aquatic model organisms to simulated microgravity [83]. The facility also contains cages for Hindlimb Unloading (HU), a widely used method of simulating weightlessness in rodents. The method is based on elevating the hindlimbs of rodents inducing a 30° head-down tilt that results in a cephalad fluid shift [84,85]. This forms a more cost-effective method of identifying biological systems affected by microgravity and can be used to study specific biological systems under microgravity [85]. Such a model has the potential to incite stress in the organism potentially impacting studies on the stress response or related systems, such as immunity. However, with proper control of environmental and physiological factors, markers of stress in HU mice have been shown to return to normal levels within a few days [85].

A study examining mice under HU for 28 days found the frequency of HSC to be significantly lower in the bone marrow when compared to controls, although this discrepancy was not observed in mice that subsequently underwent 28 days of hindlimb reloading (HR), indicating this decline recovers after a short-term [71]. In contrast, the neutrophil count in the bone marrow of HU mice was increased and, after 28 days of HR, showed only a slight decrease, demonstrating that increased granulopoiesis does not completely recover in the short term. Examination of NK cells found their frequency to be significantly reduced in both peripheral blood and bone marrow in HU mice, with levels failing to return to normal after 28 days recovery, results consistent with those the authors had obtained from samples taken from astronauts [71]. Frequency of erythrocyte precursors was significantly decreased in the bone marrow of HU mice compared to controls but, unlike the NK cells, was unchanged after 28 days of HR, showing that the decline in hematopoiesis recovers over a short term. Peripheral blood of HU mice also showed decreased counts of white blood cells and lymphocytes in comparison to controls, but this was shown to recover quickly, with no difference being seen after seven days of HR. The increase in neutrophils, coupled with a decrease in lymphocytes, as reported by Cao et al., would infer an elevated NLR which concurs not only with the in vitro experiments of Paul et al. but also with their in vivo experiments that also were conducted on HU mice [71,80]. Analysis of the HU mice in Cao et al. showed that, regardless of reloading, HSC population was elevated in any mice that underwent HU [71]. However, the proliferation of these cells was not seen to have increased, indicating an inhibition of apoptosis. Further analysis of the HSC showed that in non-HR mice, the HSCs were found to have severely impaired functionality. The declines in hematopoiesis and white blood cell count, in conjunction with the impairments of HSC function and differentiation, are factors that could contribute to immune dysfunction. As an in vivo experiment, this study provides great insight into the potential mechanisms of the impact of microgravity on immune condition.

## 3. Modelling the Bone Marrow Stem Cell Niche

The majority of spaceflight-related studies on innate immunity have used relatively simple models, such as 2D-cultured primary cells in monoculture, that are unable to fully capture the cellular and structural complexity of the bone marrow stem cell niche. The variance in results reported by studies in this field may stem from reinforcing a need for more physiologically relevant bone marrow models. By replicating the spatial architecture of the cells and extracellular matrix found in vivo, 3D cell cultures improve upon 2D cell cultures, thereby reintroducing mechanical, structural, and molecular cues lost by culturing cells on flat surfaces [86]. By addressing limitations of 2D cell cultures, they are accordingly more physiologically relevant, but at the cost of increased complexity [87].

### 3.1. The Bone Marrow Microenvironment

Hematopoiesis is defined as the proliferation and differentiation of stem cells into all cellular components of the blood [88]. This process occurs within the bone marrow, specifically inside the trabeculae, an inorganic scaffold primarily composed of hydroxyapatite [89,90]. Trabecular bone is characterized by its highly porous architecture, having a porosity of 50 to 90 percent [91]. Trabecular bone forms a system of interconnecting plates and bars called the trabeculae and the bone marrow fills the resulting pores in between, creating the characteristic honeycomb appearance [92]. Within the trabeculae, HSC can be found in two types of bone marrow stem cell niches that differ in their location and cellular composition (see Figure 4). The perivascular niche is characterized by HSC in close association with sinusoidal vessels, in contact with endothelial cells of the sinusoidal endothelium [88,93,94,95]. In contrast, the endosteal stem cell niche is close to the bone surface where HSC are in contact with osteoblasts [88,93,94]. While the relation between the two niches is still under investigation, it has been suggested that the endosteal niche maintains an environment for HSC progenitor cells while the perivascular niche is a site of proliferation and further differentiation [88,93]. As both endothelial cells and osteoblasts are derived from mesenchymal stromal progenitor cells (MSC), HSCs are in contact with stromal cells in either type of bone marrow stem cell niche. Accordingly, MSCs have been shown to form a critical component of the stem cell niche and contribute to maintenance of the HSC [96,97,98]. Other cell populations of the HSC niche include adipocytes, osteoclasts, Glial cells, and endosteal macrophages, among others [99]. The extracellular matrix (ECM) of the bone marrow, comprised of fibronectin, collagen (Type I–X)s, and laminin, among other proteins, is also known to support HSC and influence their function through serving as a mechanical scaffold and the providing of growth factors [100].

Thus, in order to study hematopoiesis under the effects of microgravity, a cellular model must incorporate critical aspects of bone marrow physiology and simultaneously be as simple as possible to ensure high reproducibility and a reliable readout. Finding the correct balance between these two competing factors poses a difficult challenge for researchers. In fact, despite the in vitro cultivation and differentiation of hematopoietic progenitor cells having long been established, and the incorporation of partner cell types into co-cultures, the expansion and maintenance of hematopoietic progenitors in the long-term continues to present a challenge to researchers [94].

### 3.2. D Models of the Bone Marrow Stem Cell Niche

While studies have already investigated the impact of simulated microgravity on HSC proliferation in monoculture using rotating wall vessels, more recent works have demonstrated the important role of MSC and their interplay with HSC in the stem cell niche [37,97,98]. A bone marrow stem cell niche model consisting of HSC plated over a confluent layer of MSC was investigated and resulted in a stratified culture with multiple layers of cells [101]. Under the layer of MSC, a population of HSC progenitors was found, while a more differentiated population of HSC formed on top, likely presenting the simplest scaffold-free 3D model of the bone marrow [101]. In this case, the surface of the MSC was observed to be the primary site of HSC proliferation, while the niche beneath the MSC layer was found to contain more progenitor-like CD34^+^CD38^−^ HSC, in contrast to the population above. This culture has limitations, however, as it does not replicate the complex 3D structure of the bone marrow, an important aspect in designing a representative in vitro model [101,102].

Interestingly, the tissue MSC originate from has been observed to play a role in the formation of the niche. MSCs derived from umbilical cord enhance HSC differentiation in contrast to MSCs derived from bone marrow, which maintain the progenitor character of HSC [103].

Closer to replicating in vivo conditions of the bone marrow stem cell niche are 3D models featuring HSC and MSC in co-culture, such as scaffold-free self-aggregating spheroids consisting only of MSC and HSC [104]. Other models have used gel droplets to encapsulate the cells and extend the culture into the third dimension [105]. Cultures grown on fabricated scaffolds have also been developed, with materials such as silk, hydroxyapatite, or collagen being used [103,106,107,108]. More technically complex models are bone marrow cultures grown on chips, exploiting the recent advents in microfluidics technology [109,110,111].

#### 3.2.1. Scaffold-Free 3D Cellular Bone Marrow Models

Scaffold-free models of the hematopoietic stem cell niche present an accessible and versatile approach to a 3D bone marrow model. Spheroids (Figure 5) are a commonly used 3D cell culture model that take advantage of the self-aggregative properties of certain cell types.

A 3D model of the bone marrow stem cell niche was developed by De Barros et al., utilizing hematopoietic stem cells and hBM-MSC in co-culture spheroids to investigate the role of active osteoblasts and hBM-MSC in controlling the migration, anchorage, and proliferation of HSC [104]. Spheroids of a maximum size of 25,000 cells were formed via self-aggregation of the cells which was facilitated by culturing them on a non-adherent surface. Typically, the most common spheroid culture methods are liquid-overlay, hanging drop, or spinner culture. After four days, the cells were found to have formed spheroids of 400 µm in diameter with a complex 3D structure resembling that of reticular cells in the bone marrow. The limit in cell number was chosen as spheroids larger than 400–500 µm develop necrotic cores due to critical O_2_ concentration [104,112,113]. De Barros et al. also developed a mixed spheroid model containing a core of osteo-induced MSC surrounded by non-induced MSC, thereby simulating the surrounding of trabecular bone by reticular cells [104]. CD34^+^ HSCs originating from either bone marrow or cord blood were then seeded onto the two types of spheroids and cultured together. Both types of HSCs migrated into both types of spheroids, but the most efficient pairing was cord blood-derived HSC seeded onto non-induced MSC spheroids. Examination of the localization of the HSC was seen in the mixed spheroids to resemble the endosteal niche, where the osteo-induced MSCs were found to potentially restrain the proliferation of the HSC. Furthermore, the spheroid model also mimicked the hypoxic environment of the HSC niche due to the gradients in nutrient and oxygen supply that exist between the outer layers and the core [114]. This model facilitates a scaffold-free 3D arrangement of cells expressing their own extracellular matrix reminiscent of the endosteal niche. However, the model does not account for the inorganic part of the bone. Spheroids are also well-suited for tissue engineering in space and microgravity environments given that cell self-aggregation and spheroid formation are facilitated by the absence of gravity drawing them to substrate. Indeed, a more recent experiment investigating the perivascular niche formed heterotypic spheroids with MSC, HSC, and endothelial cells using magnetic levitation [115]. The principle of self-aggregation was also demonstrated to be true in space, with endothelial cells forming spheroids aboard the International Space Station (ISS) as part of the ESA-SPHEROIDS project and, as shown in Figure 6, will be investigated further by the SHAPE project of the DLR [116].

#### 3.2.2. Scaffold-Based 3D Cellular Bone Marrow Models

Given the extremely porous nature of the bone marrow, many scaffolds have aimed at replicating this structure, such as in the bioreactor-based culture of Braccini et al. [117]. Their perfusion culture, with a special emphasis on fluid-flow, was built upon porous disks of hydroxyapatite to simulate the inorganic bone. The scaffolds, subsequently perfused with human bone-marrow stromal cells, resulted in a system supporting co-culture with CD45^+^ hematopoietic progenitor cells. The relative proportions of the cell types could even be regulated upon supplementation with the appropriate medium, a trait that was unable to be replicated in 2D culture. The constructs, upon ectopic implantation into mice, generated bone tissue superior to constructs loaded with the same cells that had been expanded in 2D. When a similar experiment was performed with mouse bone marrow cells, the resulting constructs were observed to maintain the HSC niche upon transplantation into mice for up to eight months [118]. In the model of Braccini et al., a focus on fluid flow was deemed important for the seeding and expansion of bone marrow cells in the scaffold, but if constructed in a microgravity environment, such fluid flow may be rendered unnecessary by the lack of sedimentation, potentially allowing a simpler form of bioreactor.

In contrast to the bone-mineral-like inorganic hydroxyapatite scaffolds, hydrogels have become an increasingly common material for simulating the HSC niche. Indeed, hydrogel mimics the extracellular matrix that composes the softer part of the bone marrow, mainly comprised of collagen and dermatopontin [119]. Hydrogel scaffolds are extremely versatile, both in their structure and in their composition. Synthetic hydrogels comprised of polyethylene glycol, polydimethylsiloxane, or other polymers have been used, as well as natural materials such as Matrigel, silk, and collagen, in addition to carboxymethylcellulose [103,106,107,108,120,121,122,123,124,125]. Collagen scaffolds have been shown to be able to support the proliferation of HSC but also be utilized to promote differentiation [103,123]. While previous models of the bone-marrow have demonstrated the ability to expand HSC, many are cytokine-driven and require regular replenishment of complex hematopoietic factors, with such cultures still subject to a gradual loss of stemness [101,108,123]. Accordingly, a model of the HSC niche that requires no supplementary cytokines and, thus, less complex maintenance, such as has been previously established in hydrogel, proves an appealing prospect for space experiments [105]. Braham et al. generated co-cultures of adipocytes, osteogenic differentiated MSC, and endothelial cells together with HSC in both alginate and Matrigel scaffolds. While both gels supported HSC proliferation, the effect was more pronounced in Matrigel, which was also able to maintain CD34^+^CD38^−^ immature hematopoietic progenitors. Within the Matrigel, colocalization of the HSC with the MSC was also observed. Notably, the addition of hematopoietic cytokines to the culture was not seen to affect the maintenance of the HSC progenitor cells. An additional benefit of this model is it being one of the few to recapitulate the hypoxic nature of the endosteal stem cell niche [126,127]. As well, the usage of Matrigel in microgravity environments has been validated in clinostat experiments [128]. However, the use of Matrigel brings with it inherent drawbacks. In fact, the reproducibility of such a model is limited by the batch-to-batch variability of Matrigel. Moreover, its murine origins preclude downstream applications in humans, thus proving unsuitable for tissue engineering [129]. The culmination of these factors are such that, in many applications, synthetic or inorganic substitutes must be used instead.

More recent iterations of hydroxyapatite scaffolds have incorporated collagen, mixing mineral with hydrogel, some of which can be 3D-printed and have been validated with human bone marrow-derived MSC (hBM-MSC) [89,120]. The composition of such scaffolds and accompanying nanostructures has been demonstrated to be closely resembling that of the human trabeculae, and the use of such a scaffold is potentially the simplest way of incorporating the inorganic phase of the bone into any model of the HSC niche [130].

While many scaffold-based models have demonstrated the ability to expand HSC, they are also often characterized by a high degree of complexity. This can stem from the generation of the scaffold, maintenance of the culture (e.g., media exchange and perfusion), or aspects of both, with these potentially being prohibitive factors when designing an experiment for the rigorous spatial constraints of orbital flight. However, it is still possible to use and adapt scaffolded culture systems for use in space. Zhang et al. successfully used a scaffold comprised of poly(d,l-lactide-coglycolide) in conjunction with hBM-MSC for short-term osteogenic and adipogenic differentiation experiments aboard the SJ-10 satellite [131]. In comparison, a simulation of long-term exposure to microgravity by random positioning machine (RPM) found a hydroxyapatite/collagen scaffold to undergo remodelling, with a slight compression and collapsing of pores when observed in contrast to scaffolds kept under 1 *g* gravity [132]. Thus, an element that must be taken into consideration is the reactivity of any scaffolds in response to microgravity and whether they can withstand such conditions at all. This would prevent a loss of science in space and aid interpretation of spaceflight data by making observations attributable to changes occurring in cells versus those in the scaffold. However, as the primary function of a scaffold is to hold the structure of a culture in place, it has been argued that in microgravity, where this poses less of a concern, scaffolds may be superfluous [133].

#### 3.2.3. Microfluidic and Organ-on-a-Chip Bone Marrow Models

Most 3D cell cultures, similar to their 2D counterparts, are static, with stagnant medium that is changed manually. The past decade, however, has seen a growing interest in “Organ-on-a-chip” cultures founded upon microfluidic technology. The flow of fluid introduces a dynamic component to the culture that mimics the active microenvironments found in vivo, such as blood circulation [134].

Previous studies have demonstrated fluid flow to increase cell seeding efficiency and viability in bone marrow cultures [135]. Studies have demonstrated MSCs to be responsive to hydrostatic pressure, shear stress, and viscosity of their environment [136,137,138,139]. Indeed, the viscosity of the bone marrow has been demonstrated to be closely tied to the shear stresses experienced by the trabecular surfaces [137,140]. Torisawa et al. devised a system where mice were implanted with a bone-inducing construct that was removed after eight weeks and maintained in vitro via microfluidics [109]. After seven days of culture, the engineered bone marrow was found to retain its morphology and exhibit no difference in cell viability in comparison to a static MSC-supported cell culture. The chip also exhibited comparable numbers and distribution of HSC and hematopoietic progenitors to freshly harvested bone marrow. In comparison to the MSC supported culture, the chip model maintained the HSC at much higher proportions, with these HSCs proving functional and self-renewing after an irradiation test on the mice. However, this model is limited by its in vivo bone engineering and murine origin, which preclude many downstream applications. Additionally, the reliance of the model on cytokines within the medium, necessitating their frequent replenishment, could cause complications in long-term studies.

In contrast to the previous model, that of Sieber et al. is of human origin and is made completely in vitro [110]. Human bone marrow-derived MSCs were pre-cultured on hydroxyapatite-coated zirconium oxide-based ceramic scaffolds to facilitate the deposition of ECM and the secretion of various factors, thereby forming a niche-like microenvironment. Cord blood-derived HSC progenitors were seeded in the scaffold thereafter. Following seeding of HSCs, the scaffolds were transferred to the microfluidic system. Analysis of the MSCs revealed the expression of stem cell factors and the formation of fibronectin-based ECM, which is important in HSC homing [141]. The niche-like environment and the progenitor state of the HSCs was maintained up to four weeks. Further analysis showed that the HSCs also retain their multi-lineage potential. The time span in this work is considerably longer than most experiments regarding hematopoiesis, validating the usage of this model in long-term experiments. The authors note that while the medium was supplemented with cytokines, addition of further cell types could potentially see the necessity for cytokines being removed. As the model only utilized a 400 µL medium reservoir, such a small footprint would ease adaptation of the system for microgravity, a process that is often a considerable challenge with other microfluidic systems [142].

More recently, Glaser et al. attempted to recapitulate both the endosteal and perivascular stem cell niches within the same model with their bone marrow-on-a-chip [111]. The model was found to generate a vascular network within a fibrin matrix, with the two niches identifiable by their respective stromal cell populations. Together, the factors from both niches were observed to be sufficient to maintain the HSC progenitors for 14 days. Notably, this model does not include a mineralized component, in contrast to many other models of the stem cell niche. However, its simplicity would enable a possible adaptation to microgravity environments.

While microfluidic systems yield great potential for modelling the structure and function of human organs, their relative complexity and the resulting infrastructure needed to support them renders adaptation for spaceflight especially challenging [142]. However, the difficulty in this process results in experimental systems that are more robust, compact, and with increasing degrees of automation that can also be used for experiments on Earth. As such, the technical knowledge gained from such an undertaking may reap long-term benefits beyond microgravity research.

### 3.3. Adaptation and Application of Bone Marrow Models in Real Microgravity

A major challenge faced by researchers comes in the adaptation of their biological models to microgravity environments. Scaffold-based models, such as those outlined in Table 1, must account for the effects of microgravity and any physical changes this may cause, as observed in the work of Avitabile et al. [132]. Gel-based scaffolds may prove easier to adapt provided they are polymerized on the ground, with the induction of polymerization in microgravity introducing another obstacle. Models utilizing perfusion and pumps, such as in the work of Braccini et al. and Glaser at al., require enclosed systems that have been downsized to adapt to the small footprints required by real microgravity platforms, such severe technical challenges being a recurring theme in the Tissue Chips in Space project [111,117,142]. In contrast, the relative simplicity of scaffold-free models, such as the spheroid in Figure 5, lend themselves to a more straightforward adaptation process for space research. This results from a lack of complex fluid-flow systems and no scaffold having to be accounted for. Additionally, the self-aggregation of spheroids has already been demonstrated extensively in both simulated and real microgravity [115,116].

## 4. Conclusions

Decades of scientific experiments in real and simulated microgravity have resulted in a wealth of evidence indicating the existence of a relationship between microgravity, bone marrow, and innate immunity. Yet, as of now, the exact mechanisms of this interplay remain unclear with many studies providing conflicting results, occasionally inciting more questions than answers. While the barrier of access to pursue space experiments continues to lower, it is still significant and simulated microgravity presents a suitable alternative.

More complex analogues of the bone marrow, such as that of Glaser et al., will be able to provide further insight into hematopoiesis and bone marrow remodelling [111]. The rapid development of microfluidic systems coupled with the breakthroughs and increase in accessibility of 3D-printing technology could potentially push forth a great advance in tissue engineering. The incorporation of partner cell types into microfluidic models in conjunction with the high reproducibility afforded by 3D-printing could address issues in sample size, but also the variability inherent in the generation of complex biological models. Of course, challenges are presented by the process of adapting such models for space, which is already being addressed by the Tissue Chips in Space initiative [142].

At present, current gaps in knowledge of innate immunity in microgravity arise from a lack of studies and from many observing conflicting results. Such deficiencies in the knowledge base could be addressed by additional experiments with simpler models in real microgravity. Many studies utilize primary cells which inherently possess a degree of variability, thus emphasizing a need for multiple donors and large sample sizes for increased statistical validity. The upcoming Spheroid Aggregation & Viability in Space (SHAPE) and IMMUNO3D projects (Figure 6) aim to address this by performing space studies with a simple 3D analogue of bone marrow, shown in Figure 5, in conjunction with a high sample size. The models used are bone marrow spheroids similar to those developed by De Barros et al., utilizing primary HSC progenitors from cord blood and hBM-MSC, but additionally incorporating hydroxyapatite beads in order to represent the inorganic phase of the bone [104]. Over 1000 spheroids will be flown on the ISS and analysed for morphological changes and differences in gene expression, providing valuable insight not only into innate immunity in microgravity, but also 3D cell culture in space, laying a foundational knowledge base for future 3D cell culture space-based experiments.

## Figures and Tables

**Figure 1 biomedicines-10-00766-f001:**
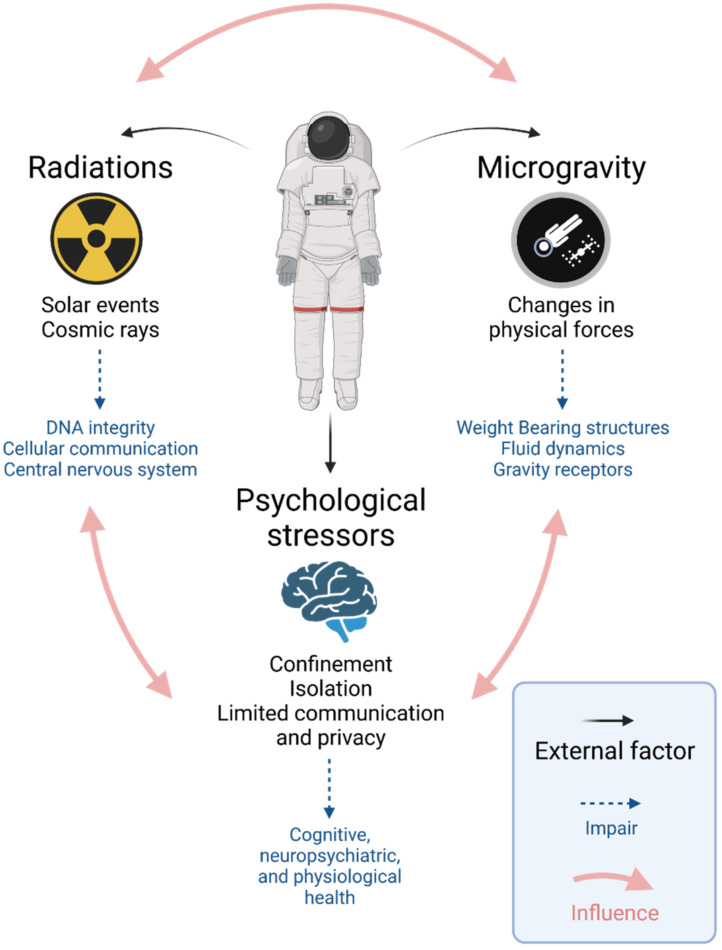
Overview of stressors and health challenges associated with spaceflight. The physical and psychological impacts of spaceflight are not restricted only to their specific stimuli but can also influence each other, resulting in a complex interplay that can have extensive health consequences for space travellers.

**Figure 2 biomedicines-10-00766-f002:**
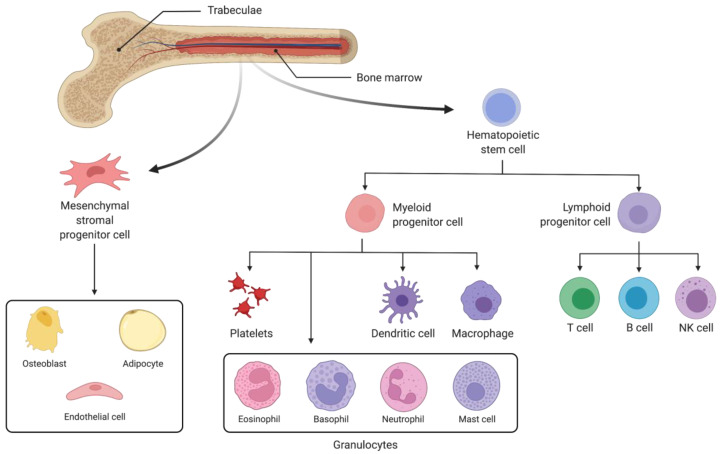
Hematopoietic stem cells and mesenchymal stem cells form the two primary cell types of the bone marrow stem cell niche. Mesenchymal cells can differentiate into a variety of cell types while hematopoietic stem cells give rise to every component of the blood including immune cells via differentiation into myeloid or lymphoid progenitors. Myeloid progenitor cells differentiate into platelets, granulocytes, or monocytes, which themselves further differentiate into dendritic cells or macrophages. Lymphoid progenitors differentiate into T cells, B cells, and NK cells. The innate immune system is comprised of NK cells, granulocytes, monocytes, macrophages, and dendritic cells, which serve as messengers between the innate and adaptive immune systems.

**Figure 3 biomedicines-10-00766-f003:**
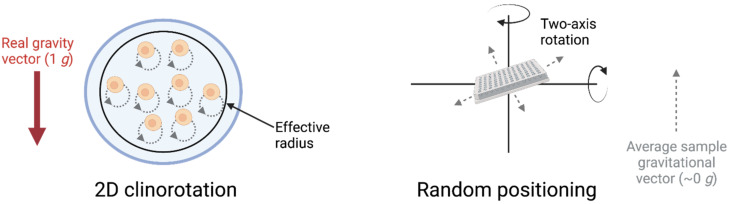
The rotation of a 2D clinostat in comparison to that of a Random Positioning Machine as platforms to simulate microgravity. The RPM rotates on two axes, operating under the principle that by rotating to numerous random positions, the gravity vector will be distributed in all directions and, over time, these will average close to a net-zero gravity vector as experienced by the sample. In contrast, clinorotation operates under a principle of constant rotation of small samples to simulate weightlessness. As distance to the centre of rotation increases, so do centrifugal forces, which may overcome thresholds of gravity sensitivity in specimens, thereby resulting in an effective radius of the simulation.

**Figure 4 biomedicines-10-00766-f004:**
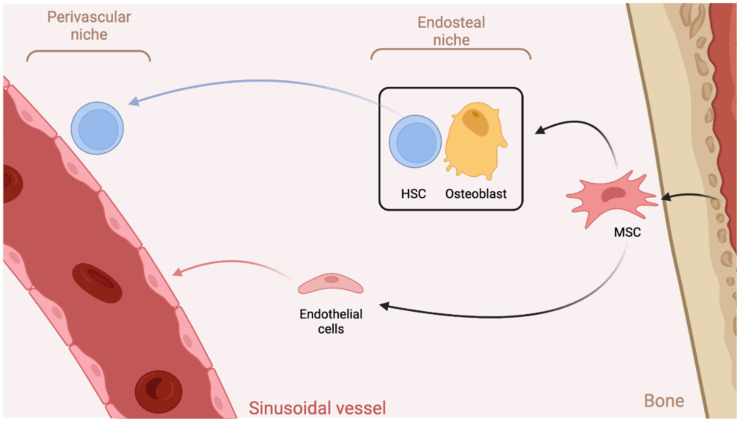
The two stem cell niches found in the bone. HSCs normally reside in either the perivascular or endosteal stem cell niches, which differ in their location and cellular composition. The perivascular niche sees HSCs in close contact with endothelial cells of sinusoidal vessels. The endosteal niche maintains HSC progenitors through populations of osteoblasts and bone marrow MSC.

**Figure 5 biomedicines-10-00766-f005:**
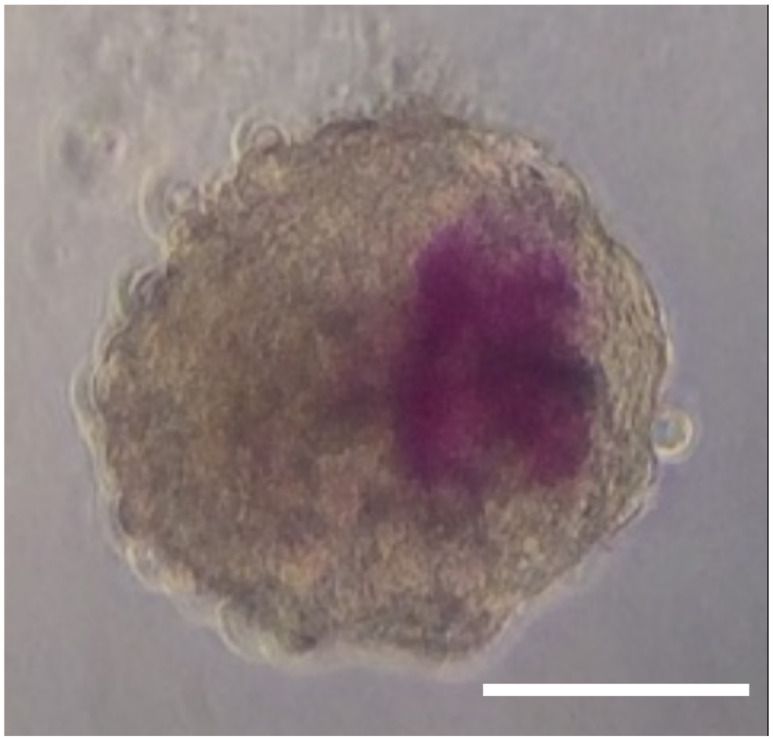
Brightfield image of a spheroid model of the bone marrow comprised of 1000 human bone marrow-derived MSC (grey, unstained) and 500 cord blood-derived HSC (purple), the latter stained with CytoTracker Orange. The spheroids are similar to those of De Barros et al. [104]. The formation of an HSC niche is evident by the concentrated distribution of HSC in the spheroid. Scale bar denotes 100 µm.

**Figure 6 biomedicines-10-00766-f006:**
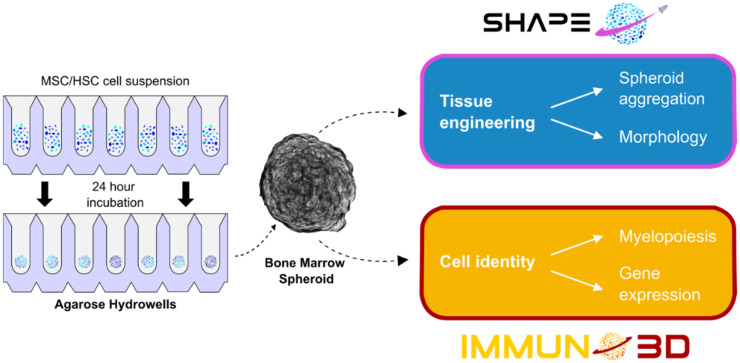
Overview of the upcoming SHAPE and IMMUNO3D projects. Both experiments use large numbers of bone marrow spheroids containing HSC and MSC generated in self-designed agarose Hydrowells in spaceflight. SHAPE is intended to lay a foundational knowledge base for spheroid cultures in space by analysing spheroid formation and morphology. IMMUNO3D aims to investigate myelopoiesis by examining the changes in gene expression, cell identity, and the changes in the proportion of the two cell types within the spheroid.

**Table 1 biomedicines-10-00766-t001:** Overview of in vitro bone marrow analogues grouped by their design philosophy.

Model Type	Description	Main Characteristics	Advantages	Limitations	Source
Scaffold-free	2D Culture	HSC plated on confluent layer of MSC	Multi-layer scaffold-free model with 2 populations of HSC	No complete 3D environment and cell–cell contacts (layers)	[101]
	Spheroid co-culture	HSC and hBM-MSC co-culture spheroids	Complex structure that resembled trabeculae	Lacks inorganic component	[104]
		Cord blood HSC seeded onto non-osteo-induced MSC spheroids	Presence of an endosteal niche	Lacks inorganic component	[114]
Scaffold-based	Mineral-based	Bioreactor perfusion culture with hBM-MSC on porous disks of hydroxyapatite	Fluid-flow integrated, implantable in mice	Bioreactor form may be unnecessary in microgravity	[118]
	Hydrogel-based	Co-culture of adipocytes, osteogenic differentiated MSC, endothelial cells and HSC in Matrigel	Proliferation of HSC while maintaining immature hematopoietic progenitor state	Batch-to-batch variability and murine origin of Matrigel, limited downstream applications	[105]
		Poly(D,L-lactide-coglycolide) hydrogel with hBM-MSC	Validated in real microgravity	Short term experiment, monoculture	[131]
	Hybrid	3D printed scaffolds of collagen and hydroxyapatite with hBM-MSC	Closely resembles human trabeculae, ease of synthesis	Not yet validated with co-culture of HSC, mechanics of 3D printing in microgravity	[120]
		hBM-MSC in HAP/collagen scaffold that underwent remodelling	Long-term exposure of scaffold to simulated microgravity	Scaffold exhibited compression and collapse of pores, RPM less constant simulation of microgravity	[132]
Micro-fluidics	In vivo bone synthesis	Development in mice then integration on chip for 7 days	HSC characteristics similar to freshly harvested bone marrow	In vivo bone synthesis phase, relies on cytokines	[109]
	In vitro bone synthesis	HAP scaffold on which MSC then HSC were seeded	Niche-like environment up to 4 weeks, small footprint	Currently relies on cytokines, frequent medium changes	[110]
	Bone marrow-on-a-chip	Both endosteal and perivascular niches maintained for 14 days	More advanced model that includes a mineralized component	Adapting microfluidics for microgravity	[111]

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
