# Peer review of "In Vitro Models of Bone Marrow Remodelling and Immune Dysfunction in Space: Present State and Future Directions"

_biomedicines, 2022, doi:10.3390/biomedicines10040766_

Round 1
Reviewer 1 Report
Dear author(s),
Dear editor,
This review article entitled “In vitro models of bone marrow remodelling and immune dysfunction in space: present state and future directions” by Sarkar & Pampaloni contains considerable information about the potential relationship between bone marrow remodelling in microgravity and its effects especially regarding hematopoietic progenitor cells. This review article highlights the microgravity driven immune dysfunctions, which cellular origins remains widely unknown. Moreover, the interactions between haematopoiesis, innate immunity and altered gravity conditions are discussed well to understand the challenges in developing in vitro models capable of delivering appropriate sample size and therefore comprehensive results. In addition, the origin and impact of the microgravity environment on hematopoietic stem cells and mesenchymal stem cells is summarized and illustrated well. The use of ground-based microgravity simulators like 2 D Clinostats, 3 D Clinostats, Random Positioning Machines (RPM) and Rotating Wall Vessels (RWV), is presented as well as spheroid formation for studies at the cellular level are well described. Knowing that each mentioned microgravity simulation platform has proven advantages and disadvantages.
On the grounds of my expertise, however, I have the following points of criticism.
Major comments
- Besides describing in detail, the proliferation and differentiation of HSC in microgravity, it would be of great value to learn more about the molecular mechanisms and signalling pathways that might be involved in this process. Which pathways are particularly affected under these circumstances? Is this directly affecting the innate immune system? Are there known feedback reactions? What are the main players of the bone-immune system cross talk?
- For this reviewer it would be a fruitful idea to shed some light on the concept of “Osteoimmunology” and specially to compare the 1g and microgravity induced adaptations.
- Within this manuscript, references are cited explaining experiments which have been done in real microgravity as well as on ground facilities, aiming to achieve comparable situations as in space, thereby allowing to perform experiments in simulated microgravity. Therefore, a short chapter 2.1. of frequently used devices was written. However, this part lacks details on the differences of the devices with respect to configuration and operation, as they might account for “often reported conflicting results” line 127. It is not correct and a contradiction to the statement in the last sentence, that these “devices generally use the same principle” line 114. The assumption that in case of the RPM, “that over time, the gravity vector averages zero” (line 114) needs a critical comment; what does this mean for a sensitive sensory system, thresholds etc.? Please comment on that!
- To what degree an object really experiences “microgravity” depends on the sensitivity of its gravity perception mechanism (please refer to: Häder, D. P., Braun, M., Grimm, D., & Hemmersbach, R. (2017). Gravireceptors in eukaryotes—A comparison of case studies on the cellular level. npj Microgravity, 3(1).
- Permanent changes in speed and direction of rotation might also present a permanent stress situation and mechanosensitive stimulation. Also refer to Hauslage et al., (Hauslage, J., Cevik, V., & Hemmersbach, R. (2017). Pyrocystis noctiluca represents an excellent bioassay for shear forces induced in ground-based microgravity simulators (clinostat and random positioning machine). npj Microgravity, 3(1), 1-7) who stated by means of a biosensor a significantly higher shear stress on the RPM compared to a 2D Clinostat.
- Understanding of potential side effects behind the simulation approach are prerequisites to reduce the risk of misinterpreting experimental results. Therefore, I recommend to add to each reference, which is cited, the mode, how microgravity was simulated. Are there examples for identical data obtained in real and simulated microgravity, in case by which method?
- Fig. 3 is quite simplified, as the effective radius is not considered, but also determines the quality of simulation. You should add 2D clinorotation.
- Within this manuscript you state later differences depending on vertical or horizontal rotation in an RWV; please explain the differential impact on the cells. Further, you state that orientation of clinorotation had an impact on the result line 141 (Chapes & Ortega); please comment on that! What does this mean? Please add the method, is it 2D clinorotation or RWV exposure?
- In case of hindlimb suspension and thus mechanical unloading of rodents you should add that this approach in animal physiology is used to study a potential influence of weightlessness on specific organ and system functions (Morey-Holton, E. R., & Globus, R. K. (1998). Hindlimb unloading of growing rats: a model for predicting skeletal changes during space flight. Bone, 22(5), 83S-88S). A critical comment with respect to stress and the immune system might be added here.
Minor comments:
- Tables and Figures should be referred in the main text of the manuscript before its first appearance
- The Graphical Abstract contains graphical parts, such as the RPM for simulated microgravity which are not specifically explained in the legend. Why has this author chosen to show only the RPM? As model for simulated microgravity?
- Whenever radiation is mentioned, I would prefer to speak of ionizing radiation. This particular space-relevant radiation consists mainly of trapped particles, heavy charged particles and high energy protons. As mentioned, also solar particle events are important.
- What is the role of Osteocytes (which account for 90-95% of all bone cells) in the process of activated macrophages which play a major role in inflammatory bone loss?
- I guess there is a flaw in numbering 3.2 -3.3 etc. Please check!
- Are the two expressions Bone marrow stem cell niche and endosteal stem cell niche handled identically within this article or are there any differences. This is not entirely clear to this reviewer.
Page 2, line 27-30: Please verify this sentence. Why are these factors preclude further exploration in space?
Page 3, line 60: What exactly is meant by the term microgravity induced bone marrow loss? Bone marrow remodelling? Which cells are involved?
Page 4, line 114-116 “these devices generally use the same principle, taking advantage of…”. On which basis could these statements be made? Observations? Please comment.
Page 4, line 121: What is meant by “democratizing microgravity” within this context?
Reviewer 2 Report
In their manuscript “in vitro models of bone marrow remodelling and immune dysfunction in space: present state and future directions”, the authors review the effects of microgravity on the bone marrow niche and cells of the innate immune system. They also discuss 3D microenvironments and their importance in studying the bone marrow nice as well as some future studies for improving the field. Overall, the review is comprehensive but requires a few modifications before publication.
- Line 73 demonstrates the need for more high throughput in vivo. ElGindi et al 2021 “Engineered Microvessel for Cell Culture in Simulated Microgravity” reported a novel platform for high throughput cellular studies under simulated microgravity.
- Line 184 – should be noted that PBMC also include lymphocytes, cells of the adaptive immune system.
- Section 2.2.2. includes studies that also look at the effects of radiation. This is the only section that combines microgravity + radiation and can be confusing to readers. There are no other studies throughout the review that consider both effects.
- Title of 3.2.3 should be “3D” ?
